# Estimation of Actual Evapotranspiration Using the Remote Sensing Method and SEBAL Algorithm: A Case Study in Ein Khosh Plain, Iran

**Amir Ghaderi** [1,2], **Mehdi Dasineh** [3], **Maryam Shokri** [4] **and John Abraham** [5,*]

1 Department of Civil Engineering, Faculty of Engineering, University of Zanjan, Zanjan 4537138791, Iran; amir_ghaderi@znu.ac.ir
2 Department of Civil Engineering, University of Calabria, 87036 Arcavacata, Rende, Italy
3 Department of Civil Engineering, Faculty of Engineering, University of Maragheh, Maragheh 8311155181, East Azerbaijan, Iran; mehdi.dasineh3180@gmail.com
4 Architecture Department, School of Engineering, University College of Nabi Akram, Tabriz 513851488, Iran; shokri.maryam1991@gmail.com
5 School of Engineering, University of St. Thomas, St. Paul, MN 55105, USA
\* Correspondence: jpabraham@stthomas.edu; Tel.: +1-612-963-2169

**Abstract:** The aim of this study was to estimate evapotranspiration (ET) using remote sensing and the Surface Energy Balance Algorithm for Land (SEBAL) in the Ilam province, Iran. Landsat 8 satellite images were used to calculate ET during the cultivation and harvesting of wheat crops. The evaluation using SEBAL, along with the FAO-Penman–Monteith method, showed that SEBAL has a sufficient accuracy for estimating ET. The values of the Root Mean Square Error (RMSE), Mean Absolute Percentage Error (MAPE), Mean Bias Error (MBE), and correlation coefficient were 0.466, 2.9%, 0.222 mm/day, and 0.97, respectively. Satellite images showed that rainfall, except for the last month of cultivation, provided the necessary water requirements and there was no requirement for the use of other water resources for irrigation, with the exception of late May and early June. The maximum ET on the Ein Khosh Plain occurred in March. The irrigation requirements showed that the Ein Khosh Plain in March, which witnessed the highest ET, did not experience any deficiency of rainfall that month. However, during April and May, with maxima of 50 and 70 mm, respectively, water was needed for irrigation. During the plant growth periods, the greatest and least amount of water required were 231.23 and 19.47 mm/hr, respectively.

**Keywords:** Evapotranspiration; water requirement; remote sensing; SEBAL; Landsat 8

## 1. Introduction

Agriculture is one of the largest draws of freshwater resources. Because of the limited water availability, agricultural sectors have been forced to increase their efficiency. One way to improve water use management and increase the efficiency is to estimate the amount of water consumed by plants and the amount involved in evapotranspiration (*ET*) [1]. Knowledge of *ET* is important for modeling hydrologic fluxes and for proper water resource management. Spatial and temporal information on *ET* not only quantifies water loss caused by evaporation, but also provides information on the relationship between land use, water allocation, and water use [2].

In addition, an optimal use of water resources will save water during times when irrigation, for instance, is not needed. Reducing the use of water in times when it is not required will not only preserve water, but will also lower the soil water content and pore pressure. This, in turn, will improve the stability of the soil and make it more resistant to landslides or other soil instabilities.

In most parts of the world, post-rainfall *ET* is the second most important element of the water cycle, so an accurate estimation at the regional scale is essential to developing appropriate management strategies [3]. Due to the limited number of meteorological stations and the high cost of data collection, satellite data is often used to provide near real-time information on meteorological and environmental parameters. An important advantage of using remote sensing to aid water consumption is that *ET* can be measured without the need to quantify other complex hydrological processes [4]. The Surface Energy Balance Algorithm for Land (SEBAL) algorithm is one remote sensing algorithm that calculates the actual *ET* based on an instantaneous energy balance at the surface of each pixel from a satellite image. This technique has been applied in many countries with success [2]. Until now, different methods and sensors have been used to estimate *ET* at regional and even global scales. The choice of method and type of sensor depend on the amount of data required, access to the sensor images, the size of the study area, and the objectives of the study.

Allen and Tasumi [5] estimated *ET* using SEBAL and Landsat satellite images in the Bear River Basin, USA. They prepared monthly *ET* maps and provided the *ET* spatial distribution. Lysimetric ground measurements were used to validate the SEBAL model data. Bastiaanssen et al. [2] estimated ET using remote sensing and the SEBAL algorithm to study water storage plans in the Yakima basin in Washington. The results show that the annual *ET* accuracy of a large basin is ~95%. Additionally, an 85% accuracy was reported on a field scale and a ~95% accuracy was reported for seasonal estimations.

Kimura et al. [6] estimated the *ET* in the Loess Plateau of China using satellite images and the SEBAL method. They provided comparisons of models and direct measurements. James et al. [7] and Jiang et al. [8] used satellite information to detect crop uniformity, vegetation percentages, and water stress and to manage irrigation systems in India, Pakistan, Sri Lanka, Argentina, and Iran. The results showed that for 85% of the cases, the parameters estimated from remote sensing corresponded to field measurements. Kosa [9] studied the effect of temperature on the actual *ET* based on Landsat satellite imagery. The results show that the relationship between the temperature and actual evapotranspiration is in the format of the polynomial equation and that this relationship can be used to estimate actual evapotranspiration when the temperature is not known.

Merlin et al. [10] pointed to the importance of *ET* for estimating water in the soil, flood forecasting, and rainfall and predicting changes in heat waves and drought. Senay et al. [11] estimated *ET* using Landsat 8 satellite images through remote sensing in the Colorado River Basin. The results showed that there were 12% and 1.3% differences between estimated and measured values for 20-day and monthly periods, respectively. Zamani Losgedaragh and Rahimzadegan [12] evaluated SEBAL, Surface Energy Balance System (SEBS), and METRIC models for estimating evaporation from a freshwater lake in Iran. The results show the SEBAL inefficiency and the proper performance of SEBS and METRIC models for estimating evaporation from the selected water body in comparison with evaporation pan data.

Elkatoury et al. [13] evaluated and compared SEBS models for estimating regional *ET* in Saudi Arabia. They showed that the monthly *ET* results measured and calculated by SEBS models were highly correlated and consistent. Faridatul et al. [14] improved remote sensing-based ET modeling in a heterogeneous urban environment. An improved surface energy balance algorithm for urban areas (uSEBAL) was proposed to make it suitable for estimating *ET* in urban environments. Finally, the results were compared with the SEBAL algorithm. Jaafar and Ahmad [15] derived a novel time-series of field-scale actual *ET* for the Bekaa Valley in Lebanon using two one-source energy balance models, utilizing local weather data and all available original Level 1 Landsat thermal imagery and Level 2 surface reflectance products. The annual analysis showed no discernable trend in *ET* across the valley, but there was an increase in irrigated agriculture in the Orontes Basin in the last five years.

Regarding the previous studies, *ET* estimates need to be evaluated and verified for each agricultural product and environmental condition. Moreover, little research has been carried out in the field on satellite estimates of *ET* for wheat crops, which is one of the most important agricultural products of Iran. Therefore, this study aims to evaluate the efficiency of the SEBAL method using Landsat 8 satellite images (with an average spatial resolution) and vegetation indices. The main objectives of this

study are (1) to evaluate spatial images of the SEBAL performance in actual *ET* validation estimation, (2) to evaluate the water requirements for the Ein Khosh Plain, and (3) to discuss the water required for irrigating the Ein Khosh Plain.

## 2. Materials and Methods

### 2.1. Study Area

As previously mentioned, the area focused on in this study is that of the Ein Khosh Plain. This is part of the Plain lands of Dehloran city in the Ilam province in western Iran. Its area is approximately 34,500 hectares and is 90 km from Dehloran city. The latitude of the region extends from 47°33′ to 47°49′ east longitude and 32°11′ to 30°25′ north latitude. This area has the longest common border between Iran and Iraq. The average height of the area is about 137 m above sea level and the average annual temperature is 25.6 °C. The average annual rainfall is 271.5 mm, and the wet season peaks in December and January. The dry period of the region is from approximately 1 April to 1 November. A map showing the region and its location in Iran and the Middle East is provided in Figure 1.

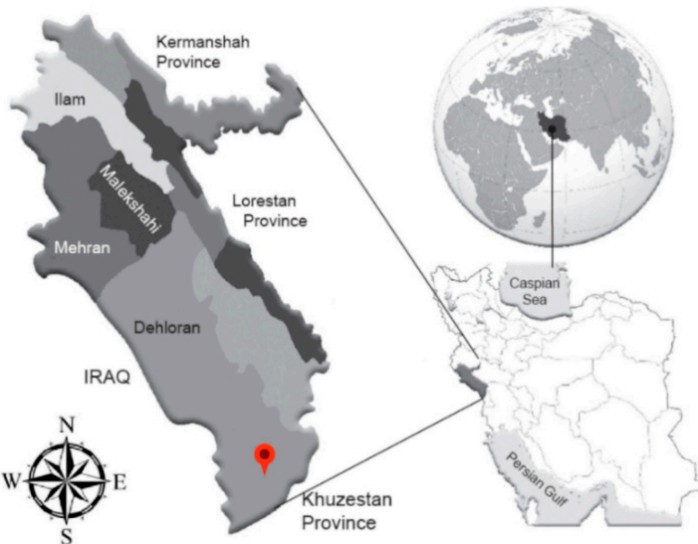

**Figure 1.** Map showing the location of the study area—Khosh Plain in the Ilam province of Iran.

### 2.2. Satellite Images

Remote sensing data obtained by satellites have the advantage of being able to provide simultaneous information over a large area. In this study, Landsat 8 satellite images processed with 11 bands were used to estimate the actual *ET* rate [16]. The 11 bands of the Landsat 8 satellite are listed in Table 1, along with wavelength and spatial resolution.

### 2.3. Calculation of Solar Radiation and ET

The Surface Energy Balance Algorithm for Land (SEBAL) was used to estimate the *ET* [17]. A conceptual schematic of SEBAL is presented in Figure 2.

Remote sensing was used to determine surface temperatures, estimates of net radiation ($R_n$), soil heat ($G$), latent heat ($\lambda ET$) fluxes, and sensible heat ($H$) in units of W/m$^2$. The latent heat flux ($\lambda ET$) represents the rate of heat loss from the surface due to *ET*, which was calculated for each pixel according to Equation (1):

$$\lambda ET = R_n - G - H \tag{1}$$

**Table 1.** Landsat 8 satellite characteristics [16].

|  | Bands | Wavelength (µm) | Spatial Resolution (m) |
|---|---|---|---|
| Landsat 8 Operational Land Imager (OLI) and Thermal Infrared Sensor (TIRS) | Band 1 – Coastal aerosol | 0.43 – 0.45 | 30 |
|  | Band 2 - Blue | 0.45 – 0.51 | 30 |
|  | Band 3 - Green | 0.53 – 0.59 | 30 |
|  | Band 4 - Red | 0.64 – 0.67 | 30 |
|  | Band 5 – Near Infrared (NIR) | 0.85 – 0.88 | 30 |
|  | Band 6 –SWIR 1 | 1.57 – 1.65 | 30 |
|  | Band 7 – SWIR 2 | 2.11 – 2.29 | 30 |
|  | Band 8 - Panchromatic | 0.50 – 0.68 | 15 |
|  | Band 9 - Cirrus | 1.36 – 1.38 | 30 |
|  | Band 10 - Thermal Infrared (TIRS) 1 | 10.60 – 11.19 | 100 |
|  | Band 11 – Thermal Infrared (TIRS) 2 | 11.50 – 12.51 | 100 |

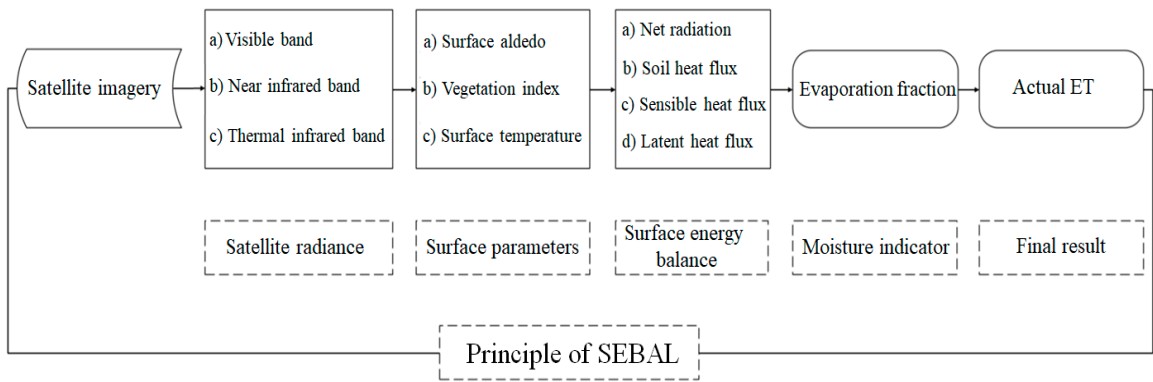

**Figure 2.** Principal components of the Surface Energy Balance Algorithm for Land (SEBAL) model [17].

Net radiation ($R_n$) is the difference between the incoming and outgoing radiative fluxes and was calculated as shown in Equation (2).

$$R_n = (1-\alpha)R_{s\downarrow} + R_{L\downarrow} - R_{L\uparrow} - (1-\varepsilon_0)R_{L\downarrow} \tag{2}$$

where $R_{s\downarrow}$ is the incoming short wavelength radiation flux, $R_{L\downarrow}$ is the incoming long wavelength radiative flux, $R_{L\uparrow}$ represents the outgoing long wave length radiative flux, $\alpha$ represents the surface albedo, and $\varepsilon_0$ is the surface emissivity. These radiant fluxes were calculated as shown in Equations (3)–(5):

$$R_{s\downarrow} = G_{sc}.\cos\theta.r.\tau_{sw} \tag{3}$$

$$R_{L\uparrow} = \varepsilon_o.\sigma.T_s^4, \tag{4}$$

$$R_{L\downarrow} = \varepsilon_\alpha.\sigma.T_\alpha^4. \tag{5}$$

Here, $G_{sc}$ is the solar constant (1367 W/m$^2$), $\cos\theta$ is the cosine of the solar incidence angle, $r$ is the Earth–Sun distance, and $\tau_{sw}$ is the atmospheric transmissivity. Values for $R_{s\downarrow}$ can range from 200 to 1000 W·m$^{-2}$, depending on the time and location of the image and on local weather conditions. The symbol $\sigma$ is the Stefan–Boltzmann constant ($5.67 \times 10^{-8}$ W·m$^{-2}$·K$^{-4}$), $T_s$ is the surface temperature (K), $\varepsilon_a$ is the atmospheric emissivity, and $T_a$ is the atmospheric temperature (K). The following empirical equation for $\varepsilon_a$ was applied using data from alfalfa fields in Idaho [18]:

$$\varepsilon_a = 0.85 \times (-Ln\tau_{sw})^{0.09}. \tag{6}$$

Here, $\tau_{sw}$ is the atmospheric transmissivity calculated assuming clear sky and relatively dry conditions. It was calculated using the elevation-based relationship of Allen et al. [19]. The soil heating

(*G*) is the rate of heat storage in the soil and vegetation due to conduction. The ratio of $G/R_n$ was computed using the following empirical equation [20]:

$$G/R_n = \frac{T_S}{\alpha}(0.0038\alpha + 0.007\alpha^2)(1 - 0.98NDVI^4),$$  (7)

where $T_s$ is the surface temperature (°C), $\alpha$ is the surface albedo, and NDVI is the normalized difference in vegetation indices between −1 and +1. Values between 0 and ~0.2 correspond to bare soil or very sparse vegetation, and NDVI > 0.2 for vegetated regions. If the NDVI value is less than zero, the surface is assumed to be water and $G/R_n = 0.5$. For areas where $T_s < 4$ °C and $\alpha > 0.45$, it is assumed to be snow-covered and $G/R_n = 0.5$ (Allen et al. [21]). NDVI was calculated from Equation (8):

$$NDVI = \frac{R' - R}{R' + R},$$  (8)

where $R$ is the reflectance in the red band and $R'$ is the reflectance in the near infrared band [20]. The sensible heat flux (*H*) is the rate of heat loss to the air by convection and conduction (Morse et al. [22]). It was obtained from Equation (9):

$$H = \frac{\rho.C_P.(T_s - T_r)}{r_a}.$$  (9)

Here, $\rho$ is the air density (kg/m$^3$), $C_p$ is the specific heat of the air at a constant pressure (1004 J·kg$^{-1}$·K$^{-1}$), $T_s$ is the surface temperature (K), $T_r$ is the air temperature at a reference level (K), and $r_a$ is the aerodynamic resistance to heat transport (s/m) (Allen et al. [19]). The term $r_a$ was computed using Equation (10):

$$r_a = 1/(C_H|V|),$$  (10)

where $C_H$ is the convective heat transfer coefficient and $V$ is the wind speed at the reference level (Tasumi et al. [23]). The term $ET_{inst}$ (instantaneous value of ET) (*J/kg*) is the ratio of λ*ET* to λ (the latent heat of vaporization) (*J/kg*) (Equation (11)):

$$ET_{inst} = 3600\frac{\lambda ET}{\lambda}.$$  (11)

Here, 3600 converts seconds to hours and λ is obtained according to Equation (12):

$$\lambda = 2.501 - (T_a - 273) \times 0.002361,$$  (12)

where $T_a$ is the atmospheric temperature (K). The $ET_{24}$ (actual daily *ET* estimation) (mm/day) is more applicable than $ET_{inst}$. SEBAL calculates $ET_{24}$ assuming that the $ET_rF$ is a 24-hour average (fixed over 24 h), according to

$$ET_{24} = ET_rF \times ET_{r-24}.$$  (13)

Here, $ET_{r-24}$ is the 24-h $ET_r$ for the day on which the image was captured; it is calculated as the sum of the hourly $ET_r$ values for that day (Allen et al. [19]). A reference value of *ET* ($ET_0$) could be obtained by using the FAO-Penman–Monteith method (Equation (14)):

$$ET_0 = \frac{\Delta(R_n - G) + \rho C_P(e_a - e_d)/r_a}{\Delta + \gamma(1 + r_c/r_a)},$$  (14)

where Δ represents the slope of the saturation vapor pressure curve (1/kPa), $\rho$ is the atmospheric density (kg/m$^3$), $C_P$ is the specific heat of the air (kJ/g°C), $e_a$−$e_d$ represents the water vapor pressure deficiency (kPa), and the terms $r_c$ and $r_a$ are the (bulk) surface and aerodynamic resistances (s/m) and $\gamma$

psychometric constants ($0.665 \times 10^{-3}$ Pa) (Allen et al. [24]). The actual annual *ET* (Equation (16)) was calculated using daily *ET* data (Equation (15)) as follows:

$$ET_{period_i} = \frac{ET_{a_i}}{ET_{o_i}} \sum_{J=k}^{I} ET_{o_j},$$ (15)

$$ET_{annual} = \sum ET_{period_i}.$$ (16)

Here, $ET_{ai}$ is the actual *ET* obtained from the images on the same day of the image being taken (*i*th day of the year) (mm), $ET_{oi}$ is the reference *ET* from the FAO-Penman–Monteith equation (also for the *i*th day of the year) (mm), $ET_{oj}$, is the ET related to the number of days in the period of image *i* that varies from the *k*th to the *l*th day of the year, and *j* represents the number of days. The last term, $ET_{annual}$, is the actual annual ET obtained from the sum of the $ET_{period}i$ (mm).

To calculate the annual *ET*, Landsat 8 satellite images were used during cropping and harvesting times and in clear sky conditions. Wheat is planted in autumn (late November) in the Ein Khosh Plain. The crop harvest occurs at the end of June. Images for this time range were obtained and ENVI-4.2 software was used to process and prepare those images for the SEBAL algorithm. In addition, REF-ET was used to calculate the reference *ET*. The dates of the images are presented in Table 2.

**Table 2.** Dates of the images used in the present study.

| Satellite | Date of Pictures (AD) |
| --- | --- |
| Landsat 8 | 11-12-2014 |
| Landsat 8 | 10-1-2015 |
| Landsat 8 | 29-2-2015 |
| Landsat 8 | 27-3-2015 |
| Landsat 8 | 20-4-2015 |
| Landsat 8 | 17-5-2015 |
| Landsat 8 | 04-6-2015 |

The statistical criteria of the Mean Bias Error (MBE), Root Mean Square Error (RMSE), Mean Absolute Percentage Error (MAPE), and correlation coefficient ($R^2$) were used to evaluate the model. These metrics were calculated as follows:

$$MBE = \frac{1}{N} \sum_{i=1}^{N} (O_i - P_i)$$ (17)

$$RMSE = \sqrt{\frac{\sum_{i=1}^{N} (O_i - P_i)^2}{N}}$$ (18)

$$MAPE = \left[ \frac{1}{N} \sum_{i=1}^{N} \left| \frac{O_i - P_i}{O_i} \right| \right] \times 100$$ (19)

Here, $O_i$ represents the observed values of the FAO-Penman–Monteith equation as the standard model; $P_i$ represents the estimated values from the SEBAL algorithm; and $\overline{O_i}$ and $\overline{P_i}$ are the mean values from the FAO Penman–Monteith model and SEBAL, respectively.

## 3. Results and Discussions

### 3.1. The Net Radiation ($R_n$), Soil Heat (G), and Sensible Heat (H) Fluxes

Figure 3 shows the net radiation ($R_n$), soil heat (G), and sensible heat (*H*) fluxes for the Ein Khosh Plain. According to the map and the $R_n$ values, the maximum radiation occurs in areas of vegetation

growth and minimum values occur in areas without vegetation. Additionally, it was observed that the values of *G* in vegetated areas were in the range of 0.05 to 0.15, which is the rate of conduction heat transfer within the soil. The sensible heat results are also plotted in Figure 3.

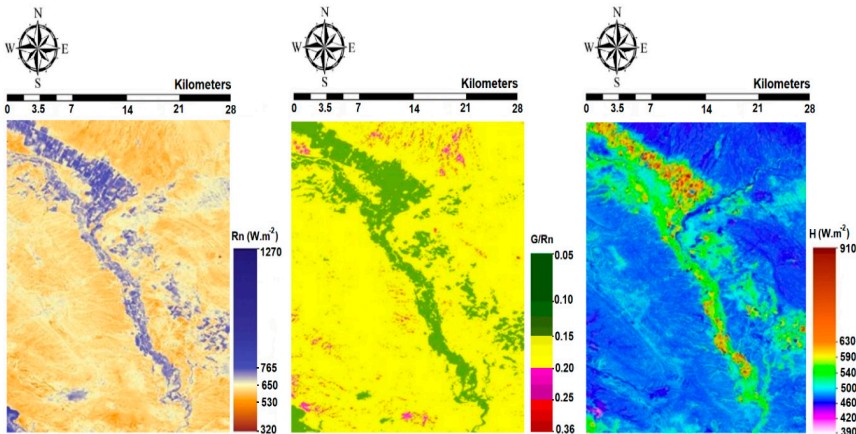

**Figure 3.** Net radiation (Rn), soil heat (G), and sensible heat (H) fluxes for the Ein Khosh Plain.

### 3.2. Evaluation of SEBAL's Performance in Actual ET Validation Estimation

After estimating $R_n$, *G*, and *H*, it was possible to determine the daily *ET* rates and the results could be compared with calculations using the FAO-Penman–Monteith equation as the reference method. The nearest station with daily and hourly data is located at 32° 15′ north and 48° 24′ east. Using 3-h station data, the daily *ET* was calculated and the water requirements were obtained during the growth period. The *ET* rates obtained from the FAO-Penman–Monteith and SEBAL methods are presented in Table 3. Furthermore, Figure 4 shows a comparison of the actual *ET* values calculated by SEBAL with the FAO-Penman–Monteith model values.

**Table 3.** Evaluation of evapotranspiration (*ET*) of FAO-Penman–Monteith and SEBAL methods.

| Date of Pictures (AD) | FAO-Penman-Monteith | SEBAL |
|---|---|---|
| | $ET_0$ (mm/day) | $ET_0$ (mm/day) |
| 11-12-2014 | 3.87 | 3.51 |
| 10-01-2015 | 4.21 | 4.53 |
| 29-02-2015 | 4.89 | 5.01 |
| 27-03-2015 | 5.73 | 5.16 |
| 20-04-2015 | 8.22 | 8.44 |
| 17-05-2015 | 9.54 | 8.98 |
| 04-06-2015 | 9.51 | 8.74 |

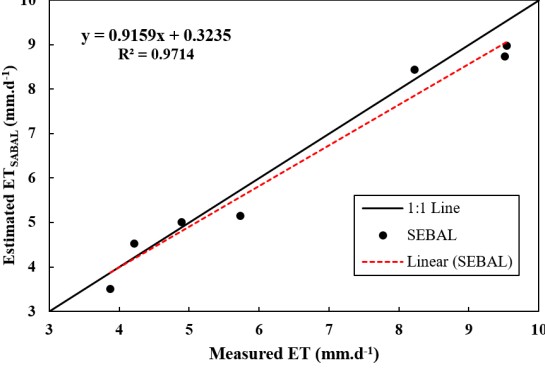

**Figure 4.** Comparison of the actual ET values calculated using SEBAL with the FAO-Penman–Monteith method.

As is shown in Table 3 and Figure 4, the values of *RMSE*, *MAPE*, and *MBE* were 0.466, 2.9%, and 0.222 mm/day, respectively, with a correlation coefficient of 0.97, indicating that SEBAL's accuracy is sufficient for estimating the actual *ET*. It can be seen that remote sensing is an efficient and effective way to estimate *ET* on large scales, especially in areas where meteorological data is not available. The actual daily *ET* rates in the Ein Khosh Plain for different months are shown in Figure 5.

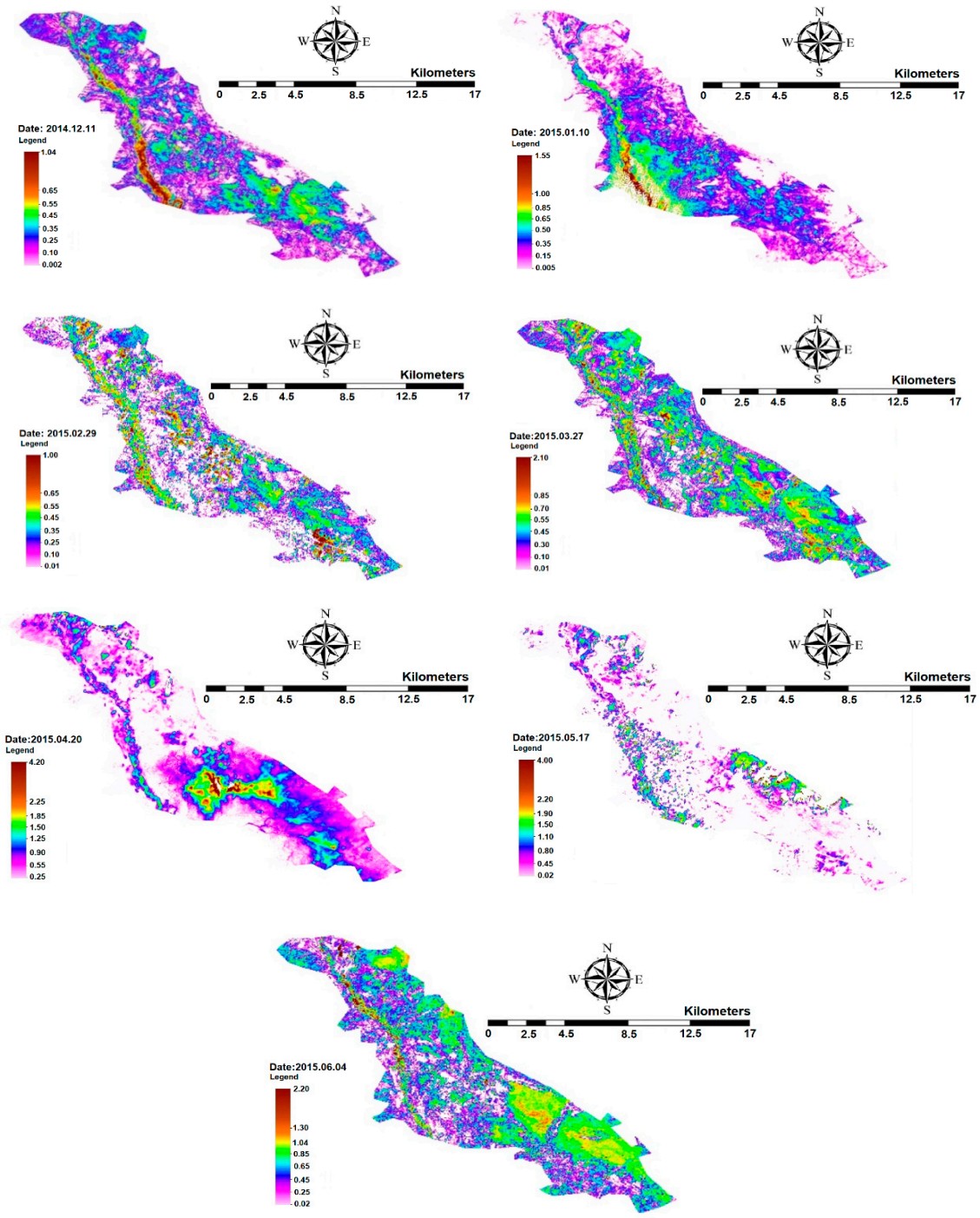

**Figure 5.** Actual daily ET rate in the Ein Khosh Plain during different months.

Figure 6 shows the actual annual *ET* in the Ein Khosh Plain. It can be observed that the maximum annual *ET* occurs in areas with a high density of vegetation.

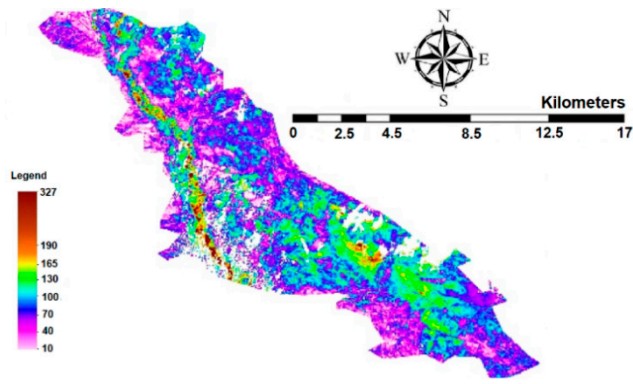

**Figure 6.** Actual annual ET in the Ein Khosh Plain (mm/year).

*3.3. Estimation of the Water Requirement for the Ein Khosh Plain*

In order to estimate the water requirement of the Ein Khosh Plain, the study areas were classified into three categories: cultivated, not cultivated/fallow, and rangeland/wasteland. Images were classified using an object-oriented method and eCognation software was employed. Each image was classified within the software based on a threshold value. Image segmentation was performed in order to subdivide the overall image into multiple non-overlapping parts (Zoleikani et al. [25]). An example outcome of this process is provided in Figure 7.

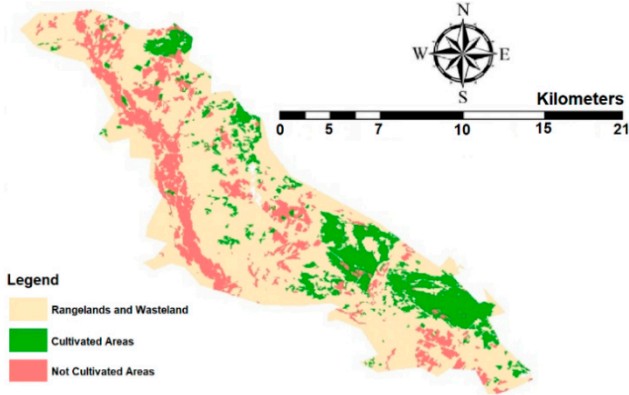

**Figure 7.** Classified image of the Ein Khosh Plain (28 May).

The area of each category of the Ein Khosh Plain is presented in Table 4.

**Table 4.** Area of use for the Ein Khosh Plain.

| Plain | Uses | Area (km$^2$) | Average Actual Annual ET |
|---|---|---|---|
| | Rangeland and wasteland | 239.99 | 75 |
| Ein Khosh | Cultivated | 62.89 | 121 |
| | Not cultivated | 60.23 | 113 |

The evaluation of *ET* in the Ein Khosh Plain shows that the 121 mm average *ET* is related to agricultural lands. The total area of the plain is 363.11 km$^2$. Taking into account the area of the Ein Khosh Plain and the cultivated land, the cultivation density is 17.21%. The normalized difference vegetation index (NDVI) was calculated according to Equation (7) and is shown in Figure 8.

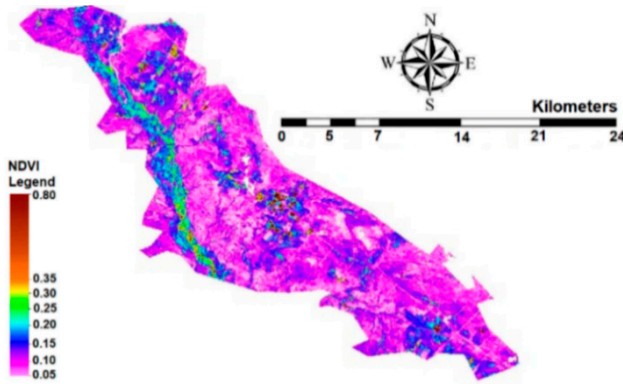

**Figure 8.** A normalized difference vegetation index (NDVI) map of the Ein Khosh Plain (28 May).

Figure 8 highlights that the NDVI values between 0.2 and 0.5 correspond to agricultural cover, representing a small area of the Ein Khosh Plain. To study water management in these periods, first, the amount of water consumed per hectare for each agricultural land was calculated. Next, the amount of water needed for irrigation was estimated based on the amount of rainfall. Table 5 shows the amount of water consumed per hectare for the Ein Khosh Plain agricultural lands.

**Table 5.** Calculation of the amount of water consumed per hectare of Ein Khosh Plain lands.

| Image Date | Study Period | Uses | Area (hr) | Amount of Water Required (mm/hr) |
|---|---|---|---|---|
| 11-12-2014 | 1st —18 November to 25 December | agricultural lands | 6318.5 | 127.11 |
| 10-01-2015 | 2nd—26 December to 15 February | agricultural lands | 6318.5 | 177.62 |
| 29-02-2015 | 3rd—16 February to 15 March | agricultural lands | 6318.5 | 19.47 |
| 27-03-2015 | 4th—16 March to 13 April | agricultural lands | 6318.5 | 231.23 |
| 20-04-2015 | 5th—14 April to 9 May | agricultural lands | 6318.5 | 227.22 |
| 17-05-2015 | 6th—10–25 May | agricultural lands | 6318.5 | 143.57 |
| 04-06-2015 | 7th—26 May to 10 June | agricultural lands | 6318.5 | 24.53 |

According to Table 5, it could be observed that the highest amount of water required was found in the fourth period (March 16 to April 13), with a value of 231.23 mm/hr, and the lowest was found in the third period (February 16 to March 15), with a value of 19.47 mm/hr, for agricultural land use.

*3.4. Water Required for Irrigation of Ein Khosh Plain in Each Period*

Figure 9 shows the monthly irrigation requirement of the Ein Khosh Plain for different months of the year.

It could be observed that the rainfall in December is higher than the water requirements and as a consequence, there is an excess of rainwater. The dark green areas correspond to surplus water in agricultural land and the value is ~15 mm. Rainfall in January exceeds *ET* and there is no need to irrigate the agricultural areas. Despite the low *ET* in February, rainfall is extremely low during this month and little irrigation water is required. The yellow and green regions correspond to areas with 0–5 and 5–20 mm water deficiencies. Additionally, March's rainfall corresponds to *ET* losses and consequently, does not require irrigation. By comparing the April irrigation requirement map with the NDVI map, the denser parts of the vegetation appear to require about 50 mm of irrigation water. Moreover, according to Figure 9 and the map of NDVI, the maximum water requirement in May is 70 mm for dense vegetation. The yellow areas on the map correspond to water deficiencies of up to 15 mm. Due to the lack of rainfall in June, the rainfall in the whole area is less than the amount required. The maximum water deficit during this month is 70 mm, which must be supplied using other water sources. Figure 10 shows the annual irrigation requirements for the Ein Khosh Plain. A careful inspection of Figure 10 and comparison with Figure 6 reveals the irrigation requirement for the dark

green parts of the map, which are deficient by up to 20 mm during the crop season. These spots are very small and not noticeable. This indicates that the land does not require irrigation. Irrigation may occur during the growing season due to a lack of rainfall, but this requirement is not seen throughout the growing season. The rainfall in the region seems to be responsive to the amount of water required in the Ein Khosh Plain.

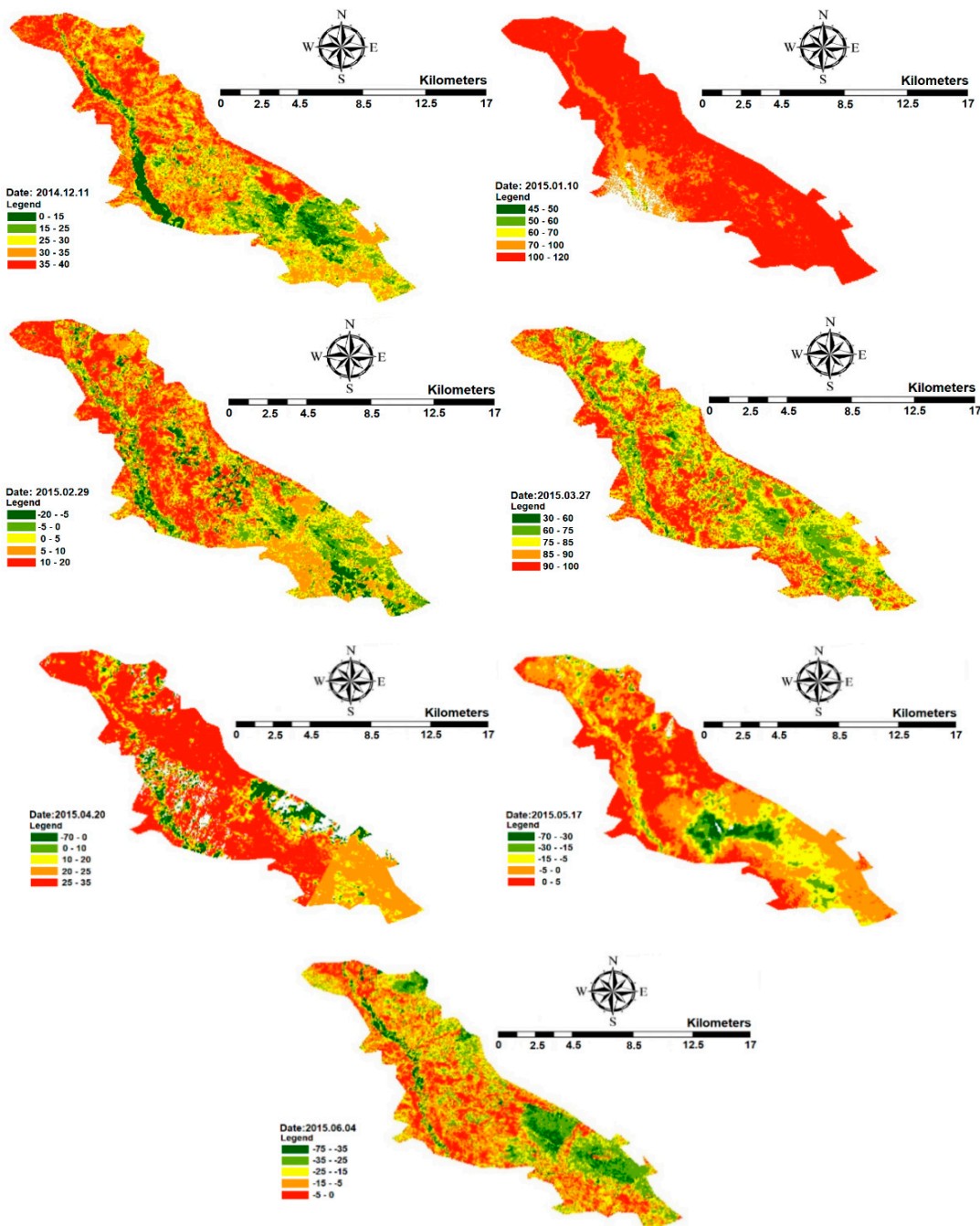

**Figure 9.** Monthly irrigation requirement of the Ein Khosh Plain for different months of the year.

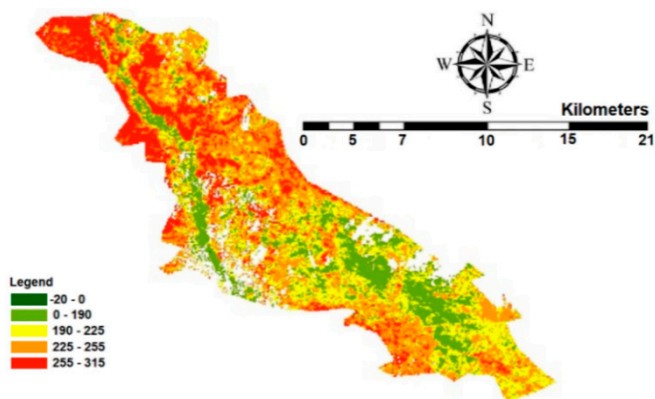

**Figure 10.** Annual irrigation requirement of the Ein Khosh plain.

## 4. Conclusions

Proper estimations of the plant *ET* and water requirements of plants are very important for improving water management and increasing the water consumption efficiency. In this regard, satellite *ET* estimation models such as SEBAL can be useful. Of course, the efficiency of this model is different in various climates and crops. Therefore, the purpose of this study was to evaluate the efficiency of the SEBAL algorithm for wheat crops, which is one of the most important agricultural products of Iran. Therefore, the *ET* rate was estimated by remote sensing and analyzed using the SEBAL algorithm for the Ein Khosh Plain in the Ilam province in Iran. The net radiation ($R_n$), soil heat ($G$), and sensible heat ($H$) fluxes and NDVI were plotted and analyzed. The evaluation of SEBAL with the FAO-Penman–Monteith method as a reference showed that the values of RMSE, MAPE, and MBE were 0.466, 2.9%, and 0.222 mm/day, respectively, with a correlation coefficient of 0.97. It was proven that SEBAL has a sufficient accuracy for estimating the actual *ET*. The results of the SEBAL algorithm are as follows:

- The rainfall rate in the Ein Khosh Plain, except for the last month of cultivation with very low rainfall, meets water-use requirements, except for late May and early June. Despite the lower *ET* rate for wheat in the last month, there is a need for irrigation during this month;
- An evaluation of irrigation requirements using monthly rainfall data showed that the Ein Khosh Plain in March (the rainfall corresponds to the *ET* rate for wheat corps), which displays the maximum *ET*, has no deficiency of rainfall. Some parts of the plain in several months, such as April and May, expect a rainfall value of up to 50 and 70 mm, respectively;
- While the total area of the plain is equal to 363.11 km$^2$, only 17.21% of the region is cultivated. Given that the average *ET* rate is 121 mm in the agricultural lands, a maximum of 20 mm of irrigation is required;
- During the wheat plant growth periods, the highest amount of water required was found in the fourth period (March 16 to April 13), with a value of 231.23 mm/hr, and the lowest was found in the third period (February 16 to March 15), with a value of 19.47 mm/hr, for agricultural land use.

In this study, the SEBAL model estimated the actual *ET* of the wheat crops with a sufficient accuracy compared with the FAO-Penman–Monteith method by using the minimum meteorological data. Overall, the results proved that the SEBAL algorithm can be an appropriate method for estimating the wheat *ET* and can be used as an efficient tool for managing water resources in wheat farms, forestry projects, etc.

Not only does the use of this technique help preserve water resources, but it also reduces water use during times when water is not needed. This guidance is important because excess water will increase the soil-water content and consequently the pore pressure. Excessive water can result in soils becoming unstable, producing landslides or other unintended consequences.

**Author Contributions:** Conceptualization, A.G., M.D., M.S., J.A., and B.C.; methodology, A.G., M.D., and M.S.; software, A.G, M.D, M.S., and B.C.; validation, A.G., M.D., and B.C.; formal analysis, A.G. and M.D.; investigation, A.G., M.D., M.S., and B.C.; writing—original draft preparation, A.G., M.D., M.S., and J.A.; writing—review and editing, A.G., M.D., and J.A.; supervision, J.A. and B.C.; project administration, A.G., J.A., and B.C. All authors have read and agreed to the published version of the manuscript.

**Funding:** This research received no external funding.

**Conflicts of Interest:** The authors declare no conflicts of interest.

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
