# Peer review of "Estimation of Actual Evapotranspiration Using the Remote Sensing Method and SEBAL Algorithm: A Case Study in Ein Khosh Plain, Iran"

_hydrology, doi:10.3390/hydrology7020036_

Round 1

Reviewer 1 Report

The authors provide an interesting study about evapotranspiration with remote sensing and with the surface energy alghorithm for land approach. This method has been compared with FAO-Penman-Monteith method, showing a good agreement. Maximum evapotranspiration has been found for March, where no rainfall deficiency has been found.
In the reviewer's opinion, the paper is interesting and well written. It can be eligible for publication if the following points are addressed

  • In the introduction, authors stress the importance of high-resolution images. Which is the difference, in meters, between low-resolution and high-resolution images? How much do they increase images resolution compared to lower resolution images?
  • In Table 1, authors present different bands, wavelengths and resolution. However, some bands look to have a lower resolution (100 meters) than other (30 meters). Is this relevant in terms of results? In other words, are lower-resolutions bands contribution important on the overall results?
  • Please use another symbol for dr in Eq. (3) since this looks to be a differential
  • In Eq. (19), MAE is introduced. As a suggestion, authors could rename this as MAPE (Mean Absolute Percentage Error), because it express a deviation from a reference value (Oi in this case) and it is also widely used for heat transfer computations (please make references to [1-3]) since it provides deviations from a reference value
  • In Table 3, the authors report that they present 7 different measurements, that follow a linear trend with good approximation (R2 = 0.9714, see Fig. 4). Are 7 measurements statistically significant for these studies? Didn't they have the possibility to achieve more data to start from?
  • In Fig. 5, actual daily ET rate is presented for different months. Doesn't spatial variability play a role? In other words, isn't the region analyzed too big to just consider only one value in the computations?
  • In Table 4, use superscript for "2" that appears in Area (km2)
  • Please reduce the numbers of bullet points in the conclusions

[1] Azizi, S., & Ahmadloo, E. (2016). Prediction of heat transfer coefficient during condensation of R134a in inclined tubes using artificial neural network. Applied Thermal Engineering, 106, 203-210.

[2] Iasiello, M., Bianco, N., Chiu, W. K. S., & Naso, V. (2020). Anisotropic convective heat transfer in open-cell metal foams: Assessment and correlations. International Journal of Heat and Mass Transfer, 154, 119682.

Author Response

Please see the attached response document.  Thank you for your time and careful consideration.

Reviewer 2 Report

Taking into account that the evapotranspiration is an important hydrological parameter and its estimation using remote sensing and meteorological data is one of the most widely used method worldwide with promising results for applications like water accounting, I consider that this article and study is very important from a scientific point of view. However, major content corrections need to be made, especially in terms of English expression. Also, the Conclusions need to be reformulated and expressed more clearly, because are the most important. Perhaps it should be mentioned in one small paragraph, the importance of this study for Iran, what it brought new or how it improves the method of water requirement estimation.

The attached file contains the corrections that I think should be made to this article.

Author Response

(The authors gave the same response as above.)

Reviewer 3 Report

see attached file

Author Response

(The authors gave the same response as above.)

Round 2

Reviewer 1 Report

In the reviewer's opinion, the paper is now eligible for publication.

Reviewer 3 Report

See attached file
